

# Radon and thoron exhalation rate, emanation factor and radioactivity risks of building materials of the Iberian Peninsula

Samuel Frutos-Puerto[1], Eduardo Pinilla-Gil[1], Eva Andrade[2,3],
Mário Reis[2,3], María José Madruga[2,3] and Conrado Miró Rodríguez[3,4]

[1] Department of Analytical Chemistry, University of Extremadura, Badajoz, Spain
[2] Laboratorio de Proteçao e Segurança Radiológica, Universidade de Lisboa, Lisboa, Portugal
[3] Centro de Ciencias e Tecnologias Nucleares, Bobadela, Portugal
[4] Department of Applied Physics, University of Extremadura, Cáceres, Spain

## ABSTRACT

Radon ($^{222}$Rn) and thoron ($^{220}$Rn) are radioactive gases emanating from geological materials. Inhalation of these gases is closely related to an increase in the probability of lung cancer if the levels are high. The majority of studies focus on radon, and the thoron is normally ignored because of its short half-life (55.6 s). However, thoron decay products can also cause a significant increase in dose. In buildings with high radon levels, the main mechanism for entry of radon is pressure-driven flow of soil gas through cracks in the floor. Both radon and thoron can also be released from building materials to the indoor atmosphere. In this work, we study the radon and thoron exhalation and emanation properties of an extended variety of common building materials manufactured in the Iberian Peninsula (Portugal and Spain) but exported and used in all countries of the world. Radon and thoron emission from samples collected in the closed chamber was measured by an active method that uses a continuous radon/thoron monitor. The correlations between exhalation rates of these gases and their parent nuclide exhalation (radium/thorium) concentrations were examined. Finally, indoor radon and thoron and the annual effective dose were calculated from radon/thoron concentrations in the closed chamber. Zircon is the material with the highest concentration values of $^{226}$Ra and $^{232}$Th and the exhalation and emanation rates. Also in the case of zircon and some granites, the annual effective dose was higher than the annual exposure limit for the general public of 1 mSv y$^{-1}$, recommended by the European regulations.

## INTRODUCTION

Radon and thoron are significant contributors to the average dose from natural background sources of radiation. They represent approximately half of the estimated dose from exposure to all natural sources of ionizing radiation (*United Nations Scientific Committee on the Effects of Atomic Radiation (UNSCEAR), 2008*).

Inhalation of these radioactive gases and their decay products can cause health risks, especially in poorly ventilated areas. Long-term exposure to high levels of radon/thoron

Corresponding author
Samuel Frutos-Puerto,
samfrutosp@unex.es

in home and working area increases risk of developing lung cancer (*World Health Organization, 1988*; *Brenner, 1994*). Radon is the second leading cause of increase of the probability of lung cancer after tobacco smoke (*World Health Organization, 2009*).

After its formation, these two radioisotopes are susceptible to escape, firstly from the grains constituting the material (known as emanation), and secondly, from the surface of the material (known as exhalation). These parameters depend, among other factors, on the half-life, consequently affecting the accumulation rate of these gaseous radioisotopes in indoor environments, and therefore, to the exposure of the human body to radiation. For radon, the half-life is 3.825 days while for thoron, just 55.6 s so, due to this difference, the effective dose from thoron and its progeny ($^{212}$Pb and $^{212}$Bi) is estimated around of 10% of that due to radon and its progeny ($^{214}$Pb and $^{214}$Bi) in indoor environments (*United Nations Scientific Committee on the Effects of Atomic Radiation (UNSCEAR), 2016*).

These factors lead to a complicated thoron measurement technique resulting in, the majority of the existing studies focus on the radon (*Petropoulos, Anagnostakis & Simopoulos, 2001*; *Stoulos, Manolopoulou & Papastefanou, 2003*; *Maged & Ashraf, 2005*; *Chen, Rahman & Atiya, 2010*; *Bavarnegin et al., 2013*; *López-Coto et al., 2014*; *Miro et al., 2014*; *Saad, Al-Awami & Hussein, 2014*; *Iwaoka et al., 2015*; *Andrade et al., 2017*; *Turhan et al., 2018*). Many of these studies also include measures of $^{40}$K, $^{226}$Ra and $^{232}$Th and risk indexes definitions trying to evaluate the radiological health hazards of these radionuclides (*Turhan & Gündüz, 2008*; *De With, De Jong & Röttger, 2014*; *Kumar et al., 2015*; *Kayakökü, Karatepe & Doğru, 2016*; *Madruga et al., 2018*) or the effective dose due to radon and its progeny (*Javied, Tufail & Asghar, 2010*).

Nevertheless, despite thoron indoor concentration is generally lower than for the radon, the $^{212}$Pb thoron progeny (half-life of 10.6 h) can accumulate to significant levels in breathable air, aggravating its inhalation risk (*World Health Organization, 2009*). Some studies (*Doi et al., 1994*; *Milić et al., 2010*; *Kudo et al., 2015*) have demonstrated that thoron concentrations can be comparable to radon and its progeny in some areas of elevated radiological risk. Furthermore, computational studies (*De With & De Jong, 2011*) taking into account factors such as the ventilation and air exchange, the building dimensions, dispersion and deposition, mitigation measures, and material properties indicates that thoron effective doses can reach the 35% of the total contribution.

Therefore, these studies demonstrate the recent and growing interest that has emerged in recent decades by the study of thoron (*Misdaq & Amghar, 2005*; *Kanse et al., 2013*; *Mehta et al., 2015*; *Jónás et al., 2016*; *Chitra et al., 2018*; *De With et al., 2018*; *Magnoni et al., 2018*; *Semwal et al., 2018*; *Prajith et al., 2019*) in building materials (*Hafez, Hussein & Rasheed, 2001*; *Sharma & Virk, 2001*; *De With, De Jong & Röttger, 2014*; *Kumar et al., 2015*) although no further studies has been reported yet focusing in the assessment of the thoron risk index in the building materials used in buildings.

Among the methods to measure both exhalation rate and emanation factor of radon and thoron isotopes in building materials, passive methods, that use solid-state nuclear track detector, accumulation chamber methods and active methods with radon/thoron monitors, can be found (*Zhang et al., 2012*).

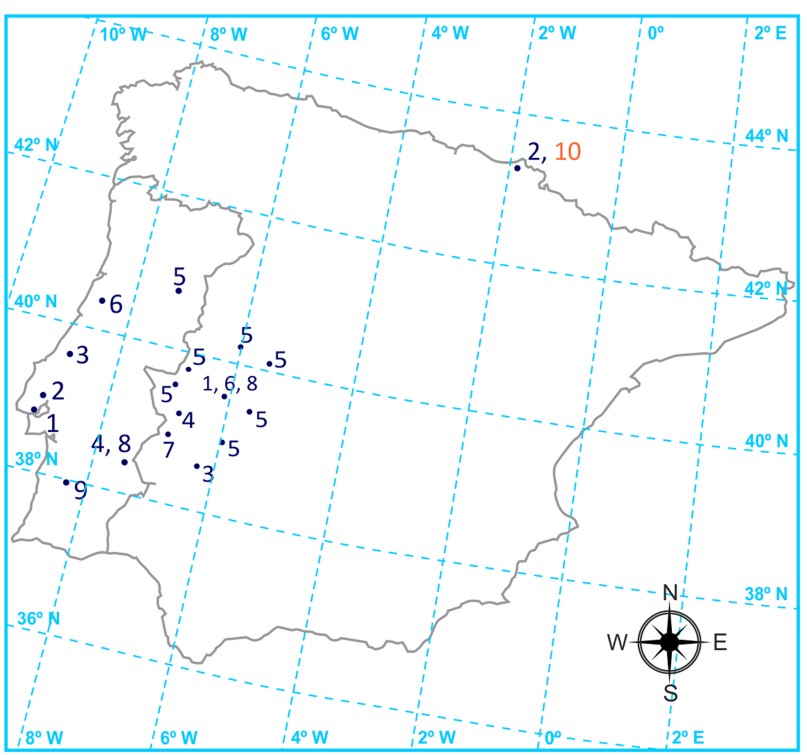

**Figure 1** Origin of the building materials. (A) NM materials: (1) Concrete, (2) Cement, (3) Marble, (4) Slate, (5) Granite, (6) Ceramic, (7) Wood, (8) Aggregate, (9) Zircon. (B) PM materials: (10) Gypsum.

In previous work, the gamma radiations emitted from $^{226}$Ra, $^{232}$Th and $^{40}$K for some of these materials were studied, as well as the radiological health hazards associated with the external gamma radiation (*Madruga et al., 2018*). In another study (*Frutos-Puerto et al., 2018*), a technique of measurement of thoron had been developed and applied to the analysis of exhalation of five materials. In the present work, expanded with more materials, we study the radon and thoron exhalation and emanation properties of an extended variety of common building materials used in the Iberian Peninsula (Portugal and Spain). The correlations between exhalation rates of these gases and their parent nuclide exhalation (radium/thorium) concentrations were examined. Furthermore, indoor radon/thoron and the annual effective dose were calculated from radon/thoron concentrations in the closed chamber. Measurements were carried out by an active method that uses a continuous radon/thoron monitor RTM1688-2 (SARAD GmbH, Dresden, Germany).

## MATERIALS AND METHODS

### Materials and sample preparation

Forty-one samples from quarries and suppliers of the most commonly used building materials manufactured in the Iberian Peninsula were collected. The mass of each sample ranged between 1 and 5 Kg. Figure 1 shows the geographical origin of the materials.

The materials were divided in two classes: materials coming from natural sources, NM, naturally occurring radioactive materials (NORM) incorporating waste after industrial processing, PM (*European Parliament, 2014*). Within each classification of materials are found:

Materials type NM:

- Concretes. Used in bulk amounts:

– Conventional
– 100% of the natural aggregate becomes electrical furnace slags
– 100% of the natural aggregate becomes blast furnace slags
– Self-compacting. High-resistance
– Mortars of resistance 5 and 7.5, respectively

- Cements. Used in bulk amounts and superficial applications:

– Type I Portland cement with less than 3% fly ash
– White cement
– Cement glue
– Rapid cement

- Natural stones. Used as bulk and superficial products:

– Marble
– Granite
– Slate

- Ceramic tiles as refractory and ceramic products to cover floors and walls, mainly:

– Tiles

- Raw materials of very different types and composition:

– Wood collected from Eucalyptus and Castahea Sativa trees
– Aggregates as sand or clay bricks
– Zircon

Materials type PM:

- Industrial products resulting from the sulfates industry of the North of Spain:

– Gypsum
– Plastic cement

Sample preparation consisted in to crushing and drying building materials in an oven for 48 h at 105 °C, prior to its grounding and sieving (2 mm particle size).

## Gamma spectroscopic analysis

To carry out the γ-emissions measurements, the milled samples were dried and placed in 160 cm³ cylindrical containers made of plastic or in 1,000 cm³ Marinelli beakers, both, hermetically sealed for 28 or more days. This period is sufficient for equilibrium to occur between the radioisotopes of $^{226}$Ra and $^{232}$Th initially contained in the material and their decay products.

To obtain the $^{232}$Th and $^{226}$Ra content an HPGe semiconductor detector was employed according to the methodology followed by *Madruga et al. (2018)*. The $^{232}$Th activity was determined by means of the γ-emissions of $^{228}$Ac (911 KeV) and $^{208}$Tl (583.01 KeV) and that of $^{226}$Ra by means of those from $^{214}$Bi (609.3 and 1764.5 KeV) and $^{214}$Pb (351.9 KeV) assuming that both radioactive series are left in secular equilibrium.

A 50% relative efficiency broad energy HPGe detector (Canberra BEGe model BE5030), with an active volume of 150 cm³ and a carbon window was used for the gamma spectrometry measurements. A lead shield with copper and tin lining shields the detector from the environmental radioactive background. Standard nuclear electronics was used and the software Genie 2000 (version 3.0) was employed for the data acquisition and spectral analysis. The detection efficiency was determined using NIST-traceable multi-gamma radioactive standards (Eckert & Ziegler Isotope Products, Berlin, Germany) with an energy range from 46.5 KeV to 1,836 KeV and customized in a water-equivalent epoxy resin matrix (density of 1.15 g cm$^{-3}$) to exactly reproduce the geometries of the samples. GESPECOR software (version 4.2) was used to correct for matrix (self-attenuation) and coincidence summing effects, as well as to calculate the efficiency transfer factors from the calibration geometry to the measurement geometry (whenever needed). The stability of the system (activity, FWHM, centroid) was checked at least once a week with a $^{152}$Eu certified point source. The acquisition time was set to 15 h and the photopeaks used for the activity determination were: 295.2 KeV (Pb-214), 351.9 KeV (Pb-214) and 609.3 KeV (Bi-214) for $^{226}$Ra; 238.6 KeV (Pb-212), 583.2 KeV (Tl-208) and 911.2 KeV (Ac-228) for $^{228}$Ra and 1,460.8 KeV for K–40. Figure 2 presents as an example a gamma-ray spectrum for a granite sample. The overall quality control of the technique is guaranteed by the accreditation of the laboratory according to the ISO/IEC 17025:2005 standards and through the participation in intercomparison exercises organized by international organizations (*Merešová, Wätjen & Altzitzoglou, 2012*; *Xhixha et al., 2017*). In summary, the activity concentration for $^{232}$Th and $^{226}$Ra ($A$) was calculated by the following expression:

$$C = \frac{N}{t\,P\,M\,\varepsilon_f} \tag{1}$$

where $N$ stands for net counts, $t$ for data collection time, $P$ for emission probability, M for mass of the sample and $\varepsilon_f$ for efficiency of the detector for the corresponding peak. Besides, uncertainty in the yield is also include since several γ-ray peaks were used for the calculation of $^{232}$Th and $^{226}$Ra activity.

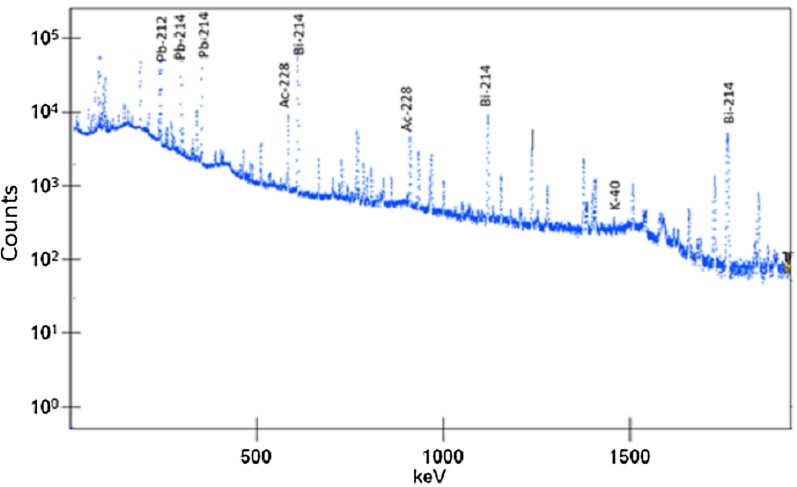

**Figure 2 Gamma-ray spectrum of a granite sample.**

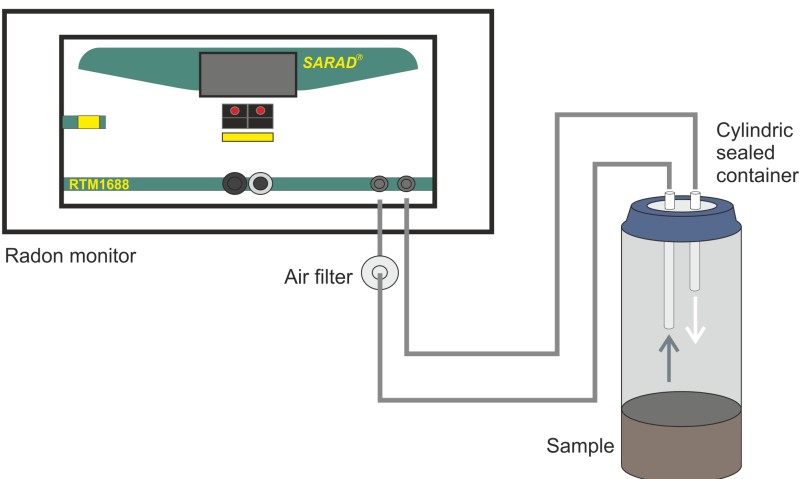

**Figure 3 Schematic experimental set-up for the radon/thoron concentration measurements.**

## Determination of massic exhalation rate and emanation factor

Exhalation is the amount of radon (radon activity) as obtained from a given layer (geological material on the surface/surface exposure) mainly the outer thinner part of the crust and it is given in Bq h$^{-1}$, according to the Netherlands Standardization Institute (*Netherlands Standardization Institute, 2001*). Exhalation can be related to the mass of the samples (massic radon/thoron exhalation, and its value is expressed Bq Kg$^{-1}$ h$^{-1}$). The method already referred (*Miro et al., 2014*; *Frutos-Puerto et al., 2018*) and similar to that of other authors (*Hassan et al., 2011*) was employed to assess the massic exhalation of $^{222}$Rn and $^{220}$Rn and it is schematized in Fig. 3.

The calculation of $^{222}$Rn and $^{220}$Rn exhalation was carried out according to the expressions presented in *Miro et al. (2014)* from the formula of the temporal variation of the radon concentration $C(t)$, in Bq m$^{-3}$:

$$\frac{dc}{dt} = \frac{EM}{V} - \lambda C - \alpha C \qquad (2)$$

where $E$ (Bq Kg$^{-1}$ h$^{-1}$) is the radon-specific exhalation rate, $M$ (Kg) the mass of the sample, $V$ (m$^3$) the air volume of the container, $\lambda$ (h$^{-1}$) the $^{222}$Rn or $^{220}$Rn decay constant and $\alpha$ (h$^{-1}$) the leakage rate from the container. The bound exhalation rate determined by hermetically closing the sample in a container can be equal to the free exhalation corresponding to the actual room conditions only in the case that the sample volume would be less than the one-tenth of the container volume. Under these circumstances, the "back diffusion" effect has no influence on exhalation rate measurements (*Krisiuk et al., 1971*). The numeric calculation are made by adjusting by least squares of the C vs t experimental data to the mathematical function given by Eq. (3). The α values obtained range approximately from 0.009 to 0.04 h$^{-1}$. For each material, such α values were considered for the calculation of the $^{222}$Rn and $^{220}$Rn exhalation.

By solving Eq. (2), the radon concentration growth as a function of time is given by:

$$C(t) = \frac{EM\left[1 - e^{-(\lambda+\alpha)t}\right]}{(\lambda + \alpha)V} + C_0 e^{-(\lambda+\alpha)t} \qquad (3)$$

being $C_0$ (Bq m$^{-3}$) the radon concentration at $t = 0$.

The $^{222}$Rn exhalation ($E_{Rn222}$) and α numeric calculation are made by adjusting by least-squares of the $C$ vs $t$ experimental data to the mathematical function given by Eq. (3).

However, due to its short half-life, after the first cycle (2 h) of measurements, the concentration of thoron in the container will reach its maximum value, remaining constant until the end of the measurements. So, from Eq. (3) the massic thoron exhalation, $E_{Rn220}$, can be calculated from the expression Eq. (4), which does not consider α value because it is much smaller than the thoron decay constant, $\lambda_{Rn220}$:

$$E_{Rn220} = \frac{C_{Rn220}\ \lambda_{Rn220}\ V}{M} \qquad (4)$$

where $C_{Rn220}$ (Bq m$^{-3}$) is the average concentration of thoron in the container during the interval of measurement from the first cycle of 2 h.

The emanation factor (amount of radon and thoron atoms that escape from the grains constituting the material into the interstitial space between the grains), $\varepsilon_{Rn}$, was calculated by the following equation for both radioisotopes (*Stoulos, Manolopoulou & Papastefanou, 2003*):

$$\varepsilon_{Rn} = \frac{E_{Rn}}{C_i \lambda_d} \qquad (5)$$

where $C_i$ is the $^{226}$Ra or $^{232}$Th content (Bq Kg$^{-1}$) of the sample for radon and thoron, respectively, $\lambda_d$, the decay constant and $E_{Rn}$ the exhalation.
Equation (5) is applicable for all measured building materials, because the dimensions of the samples were chosen to be equal to the diffusion length of these gases for these materials, around 4 cm (*Stoulos, Manolopoulou & Papastefanou, 2003*).

## Determination of annual effective dose

The $^{222}$Rn/$^{220}$Rn content accumulates in the surrounding air in a dwelling room, from building materials, depends on factors such as the room dimension, the parent element concentration, the subsequent exhalation directly from the soil and building materials in walls or soil (radon gain), the air exchange and the isotope radioactive decay. Therefore, building materials may cause an excess in the indoor $^{222}$Rn or $^{220}$Rn activity concentrations, which is described by the following equation (*Amin, 2015*):

$$A_{Rn} = \frac{E_A \, S}{V_r \, \lambda_v} \tag{6}$$

where, $A_{Rn}$, is the $^{222}$Rn or $^{220}$Rn activity concentration (Bq m$^{-3}$) in the air of the room; $E_A$ is the surface exhalation rate (Bq m$^{-2}$ h$^{-1}$); $S$ is the exhalation area (m$^2$); $V_r$ is the volume of the room (m$^3$) and $\lambda_v$ is the ventilation rate of the room (h$^{-1}$). Ratio S/V is taken to be 2 and $\lambda_v$, 0.5 h$^{-1}$ (*United Nations Scientific Committee on the Effects of Atomic Radiation (UNSCEAR), 2016*). Considering the value of the sample emanation surface in the container (0.0078 m$^2$; circumference of 5 cm$^2$), and the mass of the sample ($M$), the surface exhalation rate ($E_A$) for the building materials can be calculated, using the following equation:

$$E_A = E_{Rn} \, \frac{M}{0.0078} \tag{7}$$

This radon concentration model can then be used to determinate the annual effective doses of $^{222}$Rn by Eq. (8), recommended by the United Nations Scientific Committee on the Effects of Atomic Radiation (*United Nations Scientific Committee on the Effects of Atomic Radiation (UNSCEAR), 2016*):

$$D_{Rn222} = A_{Rn222} \, F_e \, T_a \, CF_{Rn222} \tag{8}$$

where $D_{Rn222}$ is the annual effective dose of $^{222}$Rn (Sv y$^{-1}$); $A_{Rn222}$ is the activity concentration for $^{222}$Rn (Bq m$^{-3}$); $CF_{Rn222}$ is the dose conversion factor for $^{222}$Rn progeny (Sv per Bq h m$^{-3}$); $F_e$ is the equilibrium factor for $^{222}$Rn and its progeny; and $T_a$ is the annual work time. The standard parameters were estimated using the RP 122 publication of EC 2002 (*European Commission, 2002*). The values of $CF_{Rn222}$ were assumed to be $9 \times 10^{-9}$ Sv per Bq h m$^{-3}$ and the $T_a$, 7,000 h y$^{-1}$. The value of $F_e$ was assumed to be 0.4 as reported in (*United Nations Scientific Committee on the Effects of Atomic Radiation (UNSCEAR), 2008*).

Similarly, for $^{220}$Rn:

$$D_{Rn220} = A_{Rn220} \, F_e \, T_a \, CF_{Rn220} \tag{9}$$

where, $D_{Rn220}$ is the annual effective dose of $^{220}$Rn (Sv y$^{-1}$); $A_{Rn220}$ is the activity concentration for $^{220}$Rn (Bq m$^{-3}$); $CF_{Rn220}$ is the dose conversion factor for $^{220}$Rn

**Table 1 Activity concentration for $^{226}$Ra, $C_{Ra}$, massic exhalation, $E_{Rn222}$, and emanation factor, $\varepsilon_{Rn222}$, for $^{222}$Rn of different building materials.**

| Building materials | | No. of samples ($E_{Rn222}$ > DL) | $C_{Ra}$ (Bq Kg$^{-1}$) | | | $E_{Rn222}$ (mBq Kg$^{-1}$ h$^{-1}$) | | | $\varepsilon_{Rn222}$ (%) | | |
|---|---|---|---|---|---|---|---|---|---|---|---|
| | | | Mean | SD | Range | Mean | SD | Range | Mean | SD | Range |
| NM | Concrete | 9 (7) | 27.0 | 31.8 | 7.6–87.3 | 12.2 | 8.7 | 4.3–29.0 | 8.9 | 6.7 | 1.5–17.6 |
| | Cement | 5 (1) | 28.2 | 25.1 | 21.5–76.6 | 21.0 | 3.9 | 18.4–23.8 | 11.2 | – | – |
| | Marble | 2 (1) | 22.8 | 25.3 | 4.9–40.7 | 26.3 | – | – | 8.6 | – | – |
| | Slate | 2 (2) | 28.7 | 0.2 | 28.6–28.9 | 16.0 | 97.4 | 10.4–21.6 | 7.4 | 3.6 | 4.9–9.9 |
| | Granite | 9 (9) | 122.2 | 52.9 | 51.0–239.1 | 70.3 | 71.4 | 20.5–221.4 | 8.5 | 8.7 | 2.0–24.9 |
| | Ceramic | 7 (1) | 126.4 | 105.8 | 49.9–335.0 | 0.7 | – | – | 0.2 | – | – |
| | Wood | 1 (0) | – | – | – | – | – | – | – | – | – |
| | Aggregate | 2 (1) | 69.9 | 39.7 | 41.8–97.9 | 162.5 | – | – | 22.0 | – | – |
| | Zircon | 2 (2) | 2070 | 14.4 | 48.7–4090.0 | 429.5 | 16.4 | 36.0–823.0 | 6.2 | 5.0 | 2.7–9.8 |
| PM | Gypsum | 2 (1) | 4.4 | 3.1 | 2.2–6.6 | 1.4 | – | – | 142.6 | – | – |

progeny ($40 \times 10^{-9}$ Sv per Bq h m$^{-3}$) and $T_a$ is the annual work time, 7,000 h y$^{-1}$ (*European Commission, 2002*). $F_e$ is the equilibrium factor for $^{220}$Rn and its progeny, 0.1 (*United Nations Scientific Committee on the Effects of Atomic Radiation (UNSCEAR), 2008*).

However, since the diffusion length of $^{220}$Rn is very short it is complex and ambiguous to calculate the internal exposure due to $^{220}$Rn exhaling from the building material. The indoor thoron concentration in air depends on the distance from the wall (*Doi et al., 1994*; *Javied, Tufail & Asghar, 2010*) as presented in the following equation:

$$A_{Rn220}(X) = \frac{E_{ARn220}}{\sqrt{\lambda_{Rn220} D_{eff}}} \exp\left(-\sqrt{\frac{\lambda_{Rn220}}{D_{eff}}}X\right) \qquad (10)$$

where, $A_{Rn220}(X)$ is the $^{220}$Rn concentration at a distance, $X$, from the wall. $E_{ARn220}$ is the $^{220}$Rn estimated surface exhalation rate by Eq. (7), $D_{ef}$ is the effective diffusion coefficient herein taken as 1.8 m$^2$h$^{-1}$ (*Javied, Tufail & Asghar, 2010*), $\lambda_{Rn220}$ is the decay constant of $^{220}$Rn, 45 h$^{-1}$.

It is reasonable to assume that the human respiratory organs are not more than 40 cm distance from the wall. Therefore, the $^{220}$Rn concentration at the distance of 40 cm calculated by Eq. (10), $A_{Rn220}$, is used to determinate the annual effective doses of $^{220}$Rn with Eq. (9).

## RESULTS

The results of activity concentration for $^{226}$Ra, $C_{Ra}$, massic exhalation, $E_{Rn222}$, and emanation factor, $\varepsilon_{Rn222}$, for $^{222}$Rn are summarized in Table 1.

In all samples, activity concentration for radium was above the detection limit (DL) except for the wood sample. In many samples, the exhalation rate was lower than the DL (because of $E_{Rn222}$ < DL) with exception of all samples of slate, granite and zircon.

**Table 2 Activity concentration for $^{232}$Th, $C_{Th}$, massic exhalation, $E_{Rn220}$, and emanation factor, $\varepsilon_{Rn220}$, for $^{220}$Rn of different building materials.**

| Building materials | | No. of samples | $C_{Th}$ (Bq Kg$^{-1}$) | | | $E_{Rn220}$ (Bq Kg$^{-1}$ h$^{-1}$) | | | $\varepsilon_{Rn220}$ (%) | | |
|---|---|---|---|---|---|---|---|---|---|---|---|
| | | | Mean | SD | Range | Mean | SD | Range | Mean | SD | Range |
| NM | Concrete | 9 | 14 | 9.8 | 3.9–35 | 6.3 | 2.4 | 1.9–10 | 1.2 | 0.6 | 0.6–2.1 |
| | Cement | 6 | 9.2 | 5.5 | 1.1–14 | 3.4 | 1.3 | 1.7–5.4 | 1.6 | 0.6 | 0.4–5.9 |
| | Marble | 2 | 2.9 | 1.4 | 1.8–3.9 | 3.5 | 0.3 | 3.3–3.8 | 3.1 | 1.3 | 2.2–4.0 |
| | Slate | 2 | 73 | 2.9 | 71–75 | 20 | 2.7 | 20–21 | 0.6 | 0.1 | 0.6–0.7 |
| | Granite | 9 | 51 | 33 | 10–124 | 31 | 46 | 2.6–144 | 1.1 | 1.4 | 0.2–4.8 |
| | Ceramic | 7 | 43 | 27 | 3.1–80 | 2.2 | 1.6 | 1.5–5.8 | 0.3 | 0.4 | 0.0–1.1 |
| | Wood | 1 | 0.6 | – | – | 78 | – | – | 29 | – | – |
| | Aggregate | 2 | 47 | 30 | 41–54 | 11 | 3.6 | 7.8–13 | 2.4 | 2.6 | 0.5–4.2 |
| | Zircon | 2 | 340 | 21 | 1.6–676 | 169 | 228 | 6.9–330 | 5.4 | 6.0 | 1.1–9.6 |
| PM | Gypsum | 1 | 1.4 | – | – | 2.7 | 0.3 | 2.5–2.9 | 4.0 | – | – |

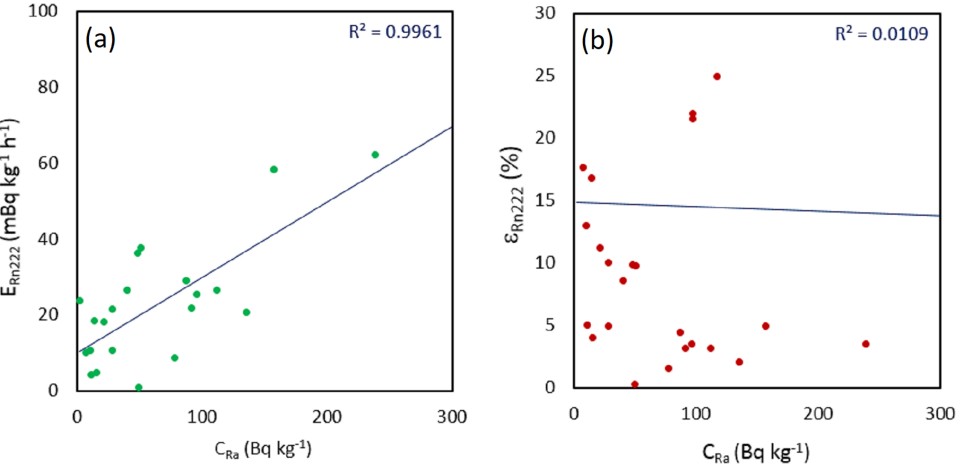

**Figure 4 Linear correlation analysis between $^{226}$Ra content and (A) $^{222}$Rn mass exhalation rate, and (B) $^{222}$Rn emanation factor.**

The maximum value on average was obtained for zircon, 429 mBq Kg$^{-1}$ h$^{-1}$, which is much higher than that found for the aggregate and the granites.

The results of activity concentration for $^{232}$Th, $C_{Th}$, massic exhalation, $E_{Rn220}$, and emanation factor, $\varepsilon_{Rn220}$, for $^{220}$Rn are summarized in Table 2.

The highest mean value for $^{232}$Th activity concentration is shown by zircon (340 Bq Kg$^{-1}$), and the lowest mean value is obtained for wood (0.6 Bq Kg$^{-1}$). The mean values of the $^{220}$Rn massic exhalation rate range from 2.2 of the ceramic to 169 Bq Kg$^{-1}$ h$^{-1}$ for zircon, respectively.

A correlation study of $^{222}$Rn mass exhalation rate with respect to $^{226}$Ra content, as shown in Fig. 4A, showed a good linear correlation coefficient ($R^2$ = 0.9961). These results show that the $^{222}$Rn mass exhalation rate increases as the $^{226}$Ra content is higher in the samples. This good linear correlation has already been observed by other authors, some

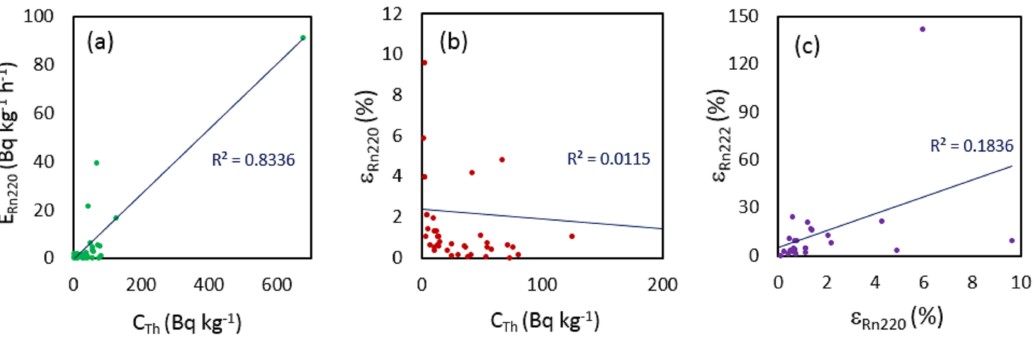

**Figure 5 Linear correlation analysis between** $^{223}$**Th content and (A)** $^{220}$**Rn mass exhalation rate, (B)** $^{220}$**Rn emanation factor. (C) Correlation analysis between the** $^{222}$**Rn and** $^{220}$**Rn emanation factors.**

**Table 3** $^{222}$**Rn surface exhalation rate,** $E_A$**, activity concentrationi in the air of the room,** $A_{Rn222}$**, and annual effective dose,** $D_{Rn222}$**, for the different building materials.**

| Building materials | | No. of samples | $E_A$ (mBq m$^{-2}$ h$^{-1}$) | | | $A_{Rn222}$ (Bq m$^{-3}$) | | | $D_{Rn222}$ (μSv y$^{-1}$) | | |
|---|---|---|---|---|---|---|---|---|---|---|---|
| | | | Mean | SD | Range | Mean | SD | Range | Mean | SD | Range |
| NM | Concrete | 9 (7) | 85 | 47 | 43–169 | 0.34 | 0.19 | 0.17–0.67 | 8.6 | 4.7 | 4.3 – 17 |
| | Cement | 5 (1) | 189 | – | – | 0.75 | – | – | 19 | – | – |
| | Marble | 2 (1) | 212 | – | – | 0.85 | – | – | 21 | – | – |
| | Slate | 2 (2) | 162 | 48 | 127–196 | 0.65 | 0.19 | 0.51–0.78 | 16 | 4.9 | 12.9 – 20 |
| | Granite | 9 (9) | 802 | 905 | 224–2843 | 3.2 | 3.6 | 0.9–11 | 81 | 91 | 23–287 |
| | Ceramic | 7 (1) | 9.2 | – | – | 0.04 | – | – | 0.9 | – | – |
| | Wood | 1 (0) | – | – | – | – | – | – | – | – | – |
| | Aggregate | 2 (1) | 1985 | – | – | 7.9 | – | – | 200 | – | – |
| | Zircon | 2 (2) | 3206 | 75 | 219–6193 | 13 | 17 | 0.9–25 | 323 | 426 | 22–624 |
| PM | Gypsum | 2 (1) | 146 | – | – | 0.58 | – | – | 15 | – | – |

of them with values very close to 1 (*Amin, 2015*). As could be expected (Fig. 4B), no correlation ($R^2 = 0.0109$) was found between the $^{222}$Rn emanation factor and the $^{226}$Ra content.

A similar correlation of $^{220}$Rn mass exhalation rate with $^{232}$Th content is shown in Fig. 5A, which shows a more weak correlation between the two quantities ($R^2 = 0.8336$). These results show that the $^{220}$Rn mass exhalation rate increases for samples with higher $^{232}$Th contents, as observed before for the $^{222}$Rn exhalation rate and $^{226}$Ra contents.

Moreover, as could be expected (Fig. 5B), no correlation ($R^2 = 0.0115$) was found between the $^{220}$Rn emanation factor and the $^{232}$Th content. Finally, no correlation ($R^2 = 0.118$) was found between the $^{222}$Rn emanation factor and the $^{220}$Rn emanation factor as shown in Fig. 5C.

The results obtained for indoor contribution, surface exhalation rate, activity concentration in the air of the room, and annual effective dose, for the different building materials had been shown in Tables 3 and 4 for $^{222}$Rn and $^{220}$Rn, respectively. Therefore,

**Table 4** $^{220}$Rn surface exhalation rate, $E_A$, activity concentration in the air of the room at 40 cm from the wall, $A_{Rn220}$, and annual effective dose, $D_{Rn220}$, for the different building materials.

| Building materials | | No. of samples | $E_A$ (Bq m$^{-2}$ h$^{-1}$) | | | $A_{Rn220}$ (Bq m$^{-3}$) | | | $D_{Rn220}$ ($\mu$Sv y$^{-1}$) | | |
|---|---|---|---|---|---|---|---|---|---|---|---|
| | | | Mean | SD | Range | Mean | SD | Range | Mean | SD | Range |
| NM | Concrete | 9 | 44 | 18 | 26–82 | 3.9 | 1.6 | 2.3–7.2 | 55 | 39 | 27–147 |
| | Cement | 5 | 22 | 6.0 | 18–32 | 2.0 | 0.5 | 1.6–2.9 | 24 | 5.8 | 19–33 |
| | Marble | 2 | 27 | 4.6 | 24–31 | 2.4 | 0.4 | 2.1–2.7 | 28 | 4.8 | 25–32 |
| | Slate | 2 | 214 | 32 | 191–236 | 19 | 2.8 | 17–21 | 220 | 32.8 | 197–243 |
| | Granite | 9 | 315 | 478 | 27–1,530 | 28 | 42 | 2.4–135 | 325 | 493 | 28–1,580 |
| | Ceramic | 7 | 24 | 13 | 17–53 | 2.1 | 1.1 | 1.5–4.7 | 25 | 13 | 18–55 |
| | Wood | 1 | 959 | – | – | 85 | – | – | 989 | – | – |
| | Aggregate | 2 | 47 | 109 | 15–80 | 4.2 | 4.1 | 1.3–7.1 | 49 | 48 | 15–83 |
| | Zircon | 2 | 1,264 | 12 | 42–2,485 | 112 | 153 | 3.7–220 | 1,300 | 1,780 | 43–2,560 |
| PM | Gypsum | 2 | 18 | 5.9 | 14–22 | 1.4 | 0.3 | 1.2 – 1.6 | 16 | 3.1 | 14–19 |

Table 3 shows that the mean values of $^{222}$Rn surface exhalation rates varied from 9.2 to 3,206 mBq m$^{-2}$ h$^{-1}$ for ceramic and zircon, respectively. The $^{222}$Rn contribution of building materials to indoor $^{222}$Rn considering the model room mentioned above, range from 0.04 for ceramic samples to 13 Bq m$^{-3}$ for zircon. As a result of this, the annual effective dose ranged from 0.9 $\mu$Sv y$^{-1}$ for ceramic to 323 $\mu$Sv y$^{-1}$ for zircon. These values are in agreement with the worldwide range (*Sola et al., 2014*; *United Nations Scientific Committee on the Effects of Atomic Radiation (UNSCEAR), 2016*).

In the case of $^{220}$Rn (see Table 4), the surface exhalation rate average varied from 22 to 1264 Bq m$^{-2}$ h$^{-1}$ for cement and zircon respectively. Its contribution of building materials to indoor $^{220}$Rn at 40 cm of the wall considering the model mentioned above, range from 2.0 for the cement to 112 Bq m$^{-3}$ for zircon. Mean values of the annual effective dose ranged from 16 $\mu$Sv y$^{-1}$ for gypsum to 1,300 $\mu$Sv y$^{-1}$ for zircon. These values are similar to those found by other authors for building materials (*Ujić et al., 2010*). However, estimation of annual effective dose from indoor thoron indicated the mean value of zircon and some values of granites had been higher than the annual exposure limit for the general public of 1 mSv y$^{-1}$, recommended by European Directive 2013/59/Euratom (*European Parliament, 2014*).

## DISCUSSION

In general, results of Table 1 are comparable to those measured in a worldwide scale (*United Nations Scientific Committee on the Effects of Atomic Radiation (UNSCEAR), 1988, 1993, 2008, 2016*; *Chen & Lin, 1997*). Thus, the values for radium content in building materials are less than the permissible value (370 Bq Kg$^{-1}$), which is acceptable as a safe limit (*OECD, 1979*). The only exception was in the radium concentration in zircon, the highest value for the mean concentration was 2,070 Bq Kg$^{-1}$. The values of exhalation rates reported in Table 1 correspond well with the values reported by other authors

(*Rawat et al., 1991*; *Porstendörfer, 1994*; *Stoulos, Manolopoulou & Papastefanou, 2003*; *Righi & Bruzzi, 2006*; *Perna et al., 2018*).

The variation in radon exhalation rates (one order of magnitude, in some cases) can be attributed to variations in radium concentrations, porosity, and surface crystallography. The emanation factor range from 0.2% to 22.0% for ceramic and aggregates respectively. These values are similar to the measured in worldwide scales (*OECD, Organization of Economic Cooperation and Development, 1979*; *United Nations Scientific Committee on the Effects of Atomic Radiation (UNSCEAR), 1993*, *2016*; *Stoulos, Manolopoulou & Papastefanou, 2003*).

The results of Table 2 show that the thoron exhalation rate is higher in zircon samples and lower in ceramic samples. This can presumably be explained by the different distributions of $^{224}$Ra parent element in the different types of samples. It should be noted how the difference among the values of exhalation rate in granites (range from 2.6 to 144 Bq Kg$^{-1}$ h$^{-1}$) reveal their different mineralogical composition. The emanation factor range from 0.3% to 29% for ceramic and wood, respectively.

The ranges of results of all these parameters are in good agreement with the values reports by other authors (*Ujić et al., 2010*; *Jónás et al., 2016*).

# CONCLUSIONS

In this study, the radon and thoron exhalation and emanation properties of building materials commonly used in the Iberian Peninsula (Portugal and Spain) were measured by using an active method with a continuous radon/thoron monitor. The correlations between exhalation rates of these gases and their parent nuclide exhalation (radium/thorium) concentrations were examined. Finally, on estimation the indoor radon/thoron, the annual effective dose was calculated.

In general, $^{226}$Ra content in building materials is less than the permissible value, 370 Bq Kg$^{-1}$, except for zircon, which means value was 2,100 Bq Kg$^{-1}$. For this material the maximum value on average of $^{222}$Rn massic exhalation rate (429 mBq Kg$^{-1}$ h$^{-1}$) was also obtained. The emanation factor $^{222}$Rn/$^{226}$Ra ranges from 0.2% to 22.0% for ceramic and aggregates, respectively. On average, the highest value for activity concentration of $^{232}$Th and massic $^{220}$Rn exhalation rate were showed by zircon, 340 Bq Kg$^{-1}$ and 169 Bq Kg$^{-1}$ h$^{-1}$, respectively. The emanation factor of $^{220}$Rn/$^{232}$Th range from 0.3% to 29% for ceramic and wood, respectively. The correlation between the radon mass exhalation rate and the $^{226}$Ra contents as well as the correlation between the thoron mass exhalation rate and $^{232}$Th contents are in good agreement.

The mean values of $^{222}$Rn surface exhalation rates varied from 9.2 to 3,206 mBq m$^{-2}$ h$^{-1}$ for ceramic and zircon, respectively. The $^{222}$Rn contribution of building materials to indoor $^{222}$Rn considering the model room mentioned above, range from 0.04 for ceramic samples to 13 Bq m$^{-3}$ for zircon. So, the annual effective dose ranged from 0.9 μSv y$^{-1}$ for ceramic to 323 μSv y$^{-1}$ for zircon.

In the case of $^{220}$Rn, the surface exhalation rate average varied from 22 to 1,264 Bq m$^{-2}$ h$^{-1}$ for cement and zircon respectively. Its contribution of building

materials to indoor $^{220}$Rn at 40 cm of the wall, range from 2.0 for cement samples to 112 Bq m$^{-3}$ for zircon. Mean values of the annual effective dose ranged from 16 μSv y$^{-1}$ for gypsum to 1,300 μSv y$^{-1}$ for zircon. Therefore, in the case of zircon and some granites, the annual effective dose was higher than the annual exposure limit for the general public of 1 mSv y$^{-1}$, recommended by the ICRP.

### Funding
This work was supported by Junta de Extremadura, Spain (projects PRI IB16114), the Air Quality Surveillance Network of Extremadura (REPICA, project 1855999FD022) and European Union Funds for Regional Development (FEDER). Eva Andrade, Mário Reis, and Maria José Madruga from the "Centro de Ciências e Tecnologias Nucleares" (C2TN) of "Instituto Superior Técnico" (IST) were supported by the Foundation for Science and Technology (FCT) in Portugal through the ID/Multi/04349/2013 project. The funders had no role in study design, data collection and analysis, decision to publish, or preparation of the manuscript.

### Grant Disclosures
The following grant information was disclosed by the authors:
Junta de Extremadura, Spain: PRI IB16114.
Air Quality Surveillance Network of Extremadura (REPICA): 1855999FD022.
European Union Funds for Regional Development (FEDER).
Foundation for Science and Technology (FCT): ID/Multi/04349/2013.

### Competing Interests
Eduardo Pinilla is an Academic Editor for PeerJ.

### Author Contributions
- Samuel Frutos-Puerto analyzed the data, prepared figures and/or tables, authored or reviewed drafts of the paper, and approved the final draft.
- Eduardo Pinilla-Gil analyzed the data, authored or reviewed drafts of the paper, and approved the final draft.
- Eva Andrade conceived and designed the experiments, performed the experiments, authored or reviewed drafts of the paper, and approved the final draft.
- Mário Reis conceived and designed the experiments, performed the experiments, authored or reviewed drafts of the paper, and approved the final draft.
- María José Madruga conceived and designed the experiments, performed the experiments, authored or reviewed drafts of the paper, and approved the final draft.
- Conrado Miró Rodríguez conceived and designed the experiments, performed the experiments, analyzed the data, prepared figures and/or tables, authored or reviewed drafts of the paper, and approved the final draft.

## Data Availability
Raw data are available as Supplemental Files.

## Supplemental Information
Supplemental information for this article can be found online at http://dx.doi.org/10.7717/peerj.10331#supplemental-information.

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
