# Peer review of "Radon and thoron exhalation rate, emanation factor and radioactivity risks of building materials of the Iberian Peninsula"

_PeerJ, doi:10.7717/peerj.10331_

## Round 0.1 · original submission · Major Revisions

Reviewers were generally complimentary of your work. Also, two reviewers have included annotated comments on attached files. Please pay attention to their comments and revise your manuscript accordingly. When you have addresses all of the reviewer comments, I would suggest that you edit the revised manuscript carefully for grammar.

Reviewer 1 ·

Basic reporting

The manuscript has been presented in a Good manner by quoting some of the references of importance. The structure of the manuscript is also acceptable as per the quality of the work.

Experimental design

The authors should specifically mention the energy of the Gamma peaks used for the assessment of Radium and Thorium in the soil samples. I would suggest that the Radium equivalent activity must be mentioned instead of Radium and Thorium and the doses should be calculated by taking into consideration the Radium equivalent.

Validity of the findings

I would like to see the graph of Gamma Spectrometry using HPGe and if possible the FWHM values as it is very important that what software is used for the curve fitting and finding FWHM values for finding the areas under the peak.

Additional comments

Figure 1 is very vague, it must be drawn wrt dome coordinates.

·

Basic reporting

The article is well written. The language of the manuscript is fluent and understandable. But there are some writting errors. These errors were corrected on the manuscript (attached as revised manuscript pdf file). The introduction section reflects the general text of the article and adequately explains why the study was done. Figures and tables are relevant and well labeled and described. However, the article was not prepared according to the rules of the journal. The authors need to rearrange the article according to the instructions for authors.

Experimental design

Research question was well defined, relevant and meaningful. Methods were described with sufficient detail. The experimental methods used in this study are very well known.

Validity of the findings

In this study, (1) the activity concentrations of 226Ra and 232Th naturally occurring in 41 building materials (structural, covering and raw materials) manufactured in the Iberian Peninsula (Portugal and Spain), exported and used in all countries of the world were measured using a gamma-ray spectrometry with an HPGe detector, (2) radon (222Rn) and thoron (220Rn) exhalation rates (mass and surface) and emanation coefficients of these materials were determined using active radon and thoron monitor, and (3) the annual effective dose due to inhalation of radon and thoron was estimated to assess the radiological risks associated the use of these building materials. These findings are important for assessing the radon and thoron concentrations in buildings, environmental radioactivity, and health hazards of individuals. For this reason, the manuscript can be considered for publication in International PeerJ.

Additional comments

But there are some writting errors. These errors were corrected on the manuscript (attached as revised manuscript pdf file).

·

Basic reporting

Specific comments
Page in MS (Line) Written Change to Comments
L 48-49 Radon is the second leading cause of increase
49 of the probability of lung cancer after tobacco smoke (Torres-Durán et al., 2014). This citation is not correct. This fact was established much before 2014. Find original reference
L.55 is 3.8 3.825
L.55 equivalent radiation dose Usually we talk about effective dose.

L62 to the detriment neglecting Sentence is too long and cumbersome
L70 progenies progeny There is not plural for progeny.
L99 materials was materials were
Eq2 Back diffusion was not taken into account in this Eq. Please explain why it was neglected
L. 167 which do not consider which does not consider
L. 219 from de wall from the wall

Experimental design

Well done

Validity of the findings

Agree with world wide

Additional comments

Major revision is need for this ms.
Experimental part of this ms is probably well done. Calibration of gamma spectrometer is missing. However, estimation of radon and thoron activity from some construction material obtained by Eq. 6 is not correct. It does not consider amount of material used in some room. So, it was obtained that effective dose (which is usually denoted with E) from zircon is larger than 1 mSv/year. My question, how much zircon can be found in some room, and how is realistic that all room is made by zircon. Formula in Eq.6 should be modified to take into account amount of material used in construction. Then, repeat calculation of effective dose In my, the best knowledge, zircon is used as coating material as very thin layer on ceramic tiles in bathroom or kitchen. Total mass of zircon is very small. It can be true or similar for some other material.

Reviewer 4 ·

Basic reporting

The paper is interesting and contains data worth to be published. However the English should be substantially improved. I’m not a native English speaker and I don’t feel qualified to do a complete linguistic revision of the text. I just corrected some major mistakes (see the attached annotated pdf). However, in my opinion, the following sentences should be rewritten to improve clarity and comprehensibility:
a) Lines 60-62
b) Lines 67-69
c) Lines 151-153
d) Lines 164-168
e) Lines 184-186 (attending ??)
f) Lines 222-224
g) Line 236 (the maximum value on average: what do you mean?)

Besides the language issue, the paper contains some problematic points, addressed more specifically in the next sections. These points need to be clarified before making the decision to publish or not

Experimental design

In equation (1) (Material and methods), the mass m of the sample is missing in the denominator: please correct
I have also some questions regarding the experimental set up described in figure 2.
a) Do you have measured (or estimated) the leakage factor  in equation (2) ? Its value is negligible for thoron but can substantially affect the radon exhalation estimation
b) Can you give the asymptotic activity concentration values (for both radon and thoron) reached in the cylindric (not cilindrical, please correct) sealed container?

Validity of the findings

My major concern regarding the findings is related to the use of equation (7) for the calculation of the surface exhalation rate from the measured mass exhalation rate. The relationship between these quantities can be highly affected by the diffusion length (l=√(D⁄lambda) ;lamba=decay constant;D=coefficiente of diffusion in the material) of radon and thoron in the different material. Moreover, the exhalation area of your experimental set up can be hardly compared to the exhalation area in real conditions. Actually, being the material crushed, you have increased the effective surface emission area compared to that of the real building material (bric, tile). This fact can affect your results.
Some other minor questions should be addressed. In the tables you take mean values for some materials (zircon, for example) with only 2 samples which show a very large difference: it seems not very informative.

Additional comments

The paper is interesting and worth publishing. I encourage the authors to answer to the above addressed points

Annotated reviews are not available for download in order to protect the identity of reviewers who chose to remain anonymous.

---

## Round 0.2 · accepted · Accept

Thank you for your efforts in responding to reviewer comments and revising your manuscript.

·

Basic reporting

Authors made necessary modifications. I do not have more comments and ms can be accepted for publication.

Experimental design

Well done and well described.

Validity of the findings

Authors found radium/radon/thoron concentration and exhalation in the range similar to other authors world wide.

Additional comments

MS can be accepted.